# Coral restoration alters reef soundscapes but machine learning and manual analyses suggest different recovery rates

Emily Maria Croasdale[1,2]*, Luca Saponari[3], Charlotte Dale[3], Nirmal Shah[3],
Ben Williams[4,5], Timothy A. C. Lamont[6]*

1 Institute for Biodiversity and Ecosystem Dynamics, University of Amsterdam, Amsterdam, The Netherlands, 2 MARE, Laboratorio Maritimo da Guia, Faculdade de Ciencias da Universidade de Lisboa, Cascais, Portugal, 3 Nature Seychelles, The Centre for Environment & Education Roche Caiman, Mahe, Republic of Seychelles, 4 Division of Biosciences, University College London, London, United Kingdom, 5 Zoological Society of London, London, United Kingdom, 6 Lancaster Environment Centre, University of Lancaster, Lancaster, United Kingdom

* emcroasdale@fc.ul.pt (EMC); tim.lamont@lancaster.ac.uk (TL)

## Abstract

Coral restoration is recognised as a critical tool to mitigate pantropical degradation of reef ecosystems. Robust monitoring of restoration progress is crucial for projects to evaluate their success, improve practice, and share knowledge. However, traditional visual surveys often fail to capture the full impact of coral restoration on reef function. Therefore, we employed Passive Acoustic Monitoring (PAM) to assess whether the soundscape of a coral restoration site in the Seychelles differs from adjacent healthy and degraded reference reefs. We applied two methods of soundscape analysis: manual detection of unidentified fish sounds; and machine learning-based Uniform Manifold Approximation and Projection analysis. Results were approach-specific: the manual approach highlighted similarities in fish calls between the restoration site and the healthy reference reef, while the machine learning approach extracted broader soundscape patterns, clustering the restoration site alongside the degraded reference reef. Although this is a single-site study, these findings suggest that a) coral restoration alters reef soundscapes, though recovery time may be taxon-specific, and b) multiple metrics are needed to bridge single-taxon and broad soundscape scales. This study contributes to the evolving field of soundscape ecology in coral reef ecosystems, highlighting the utility of PAM in monitoring changes to reef function through coral restoration.

## Introduction

Over the past decade, coral restoration efforts around the world have dramatically increased as part of an attempt to mitigate global degradation of reef habitat [1]. Despite increased effort to restore coral ecosystems, very few programmes have

**Data availability statement:** All relevant data for this study are publicly available from the Zenodo repository (https://doi.org/10.5281/zenodo.18631770).

**Funding:** Sources of funding for this project include the Adaptation Fund Climate Innovation Accelerator, awarded to Nature Seychelles (LS, CD, NS) for project PIMS 5736 (undp.org/mauritiusseychelles/projects/restoring-marine-ecosystem-services-rehabilitating-coral-reefs-meetchanging-climate-future), a Madeleine Julie Vervoort Fonds travel grant from the Amsterdams Universiteitsfonds (auf.nl) awarded to EMC, and a research fellowship from the 1851 Royal Commission, awarded to TACL (royalcommission1851.org). Funders played no role in any part of the study or manuscript preparation.

**Competing interests:** The authors have declared that no competing interests exist.

comprehensive monitoring efforts that capture the impact of coral restoration on reef ecosystem functioning [2–4]. Documenting the biological community shift that happens as a reef goes through a restoration process can inform future projects and help to evidence the effectiveness of active restoration as a tool to conserve reef-associated fauna. However, restoration projects frequently rely exclusively on traditional metrics such as coral cover and fish abundance for their impact monitoring [5]. These metrics fail to capture the full range of changes to the reef ecosystem [1,2,5].

Recently, passive acoustic monitoring (PAM) has increasingly been employed in studies on coral reef ecology, for example to establish correlations between coral assemblages and different aspects of the soundscape [6,7], to develop eco-acoustic indicators of biodiversity [8], and to further our understanding of the role of reef soundscapes as a driver of coral and fish settlement [9,10]. PAM complements traditional visual survey methods by (i) extending sampling periods to capture temporal variability, (ii) recording at night when visual surveys are not possible, (iii) removing the need for in-water observers therefore eliminating the effects of diver presence, and (iv) facilitating detection of some visually cryptic species [11–13]. Moreover, in-water acoustic sampling is often less labour-intensive than visual surveys, and recording equipment is rapidly becoming more accessible and affordable [14].

Soundscapes are a useful indicator of reef functioning, due to the role of sound production in several of the ecological processes that constitute reef function, namely: bioerosion, herbivory, predation, and secondary production [15]. Passive sounds are often produced during feeding, which contributes to bioerosion (coral-livores) and herbivory, as a result of feeding technique [13]. Further, many species of reef fishes actively produce sound during courtship, territorial, and anti-predator interactions [13,16,17], mediating the processes of herbivory, predation, and secondary production. The soundscape in turn contributes to reef function via secondary production, as one of the drivers of population recruitment for some larval-phase organisms, which use auditory settlement cues to detect and choose habitat [17–23]. Therefore, monitoring soundscape recovery provides an indication of changes in reef function, which has garnered interest and early adoption of PAM as a tool for monitoring reef health and conservation [24–26].

The biophonic soundscape may be broadly categorised into a high frequency bandwidth (>1kHz), largely dominated by invertebrates, and a low frequency bandwidth (0-1kHz) that is predominantly fish vocalisations [13]. Soundscape activity and complexity in the lower frequency bandwidth is strongly correlated with metrics associated with reef function such as high coral cover [12,27,28], benthic complexity [6], and fish biomass or diversity [28,29]. Soundscape variability is associated with seasonal, lunar, and diel cycles, though the relative strength of each cycle differs between reefs [30,31]. Diel cycles in soundscape variability are associated with peaks in activity around crepuscular periods [32], and genus-specific activity in response to moonlight [33]. Healthy reefs dominated by different coral genera develop different nighttime soundscapes in terms of sound diversity and amplitude [34]. Recent work has demonstrated that manual detection and quantification of unidentified fish sounds is a simple and ecologically relevant metric to distinguish between reefs with varying levels of coral cover [11,27].

In recent years, soundscape analysis has been performed with increasing levels of automation. This evolution is being driven in part by the extensive amount of data produced as long-term recording hardware has become increasingly affordable [14]. The advent of machine learning has aided progress in soundscape analysis, in both terrestrial [35,36] and marine environments [37–39]. In particular, pre-trained convolutional neural networks (P-CNNs) have emerged as valuable tools to extract high quality feature embedding representations from acoustic data [39,40,41]. Machine learning driven techniques pose several advantages that complement manual analyses: for example, rapid processing of large amounts of data to high resolution (< 1 second) allows identification of anomalous sounds and quantification of habitat quality [40], with reduced time demand from observers [42]. These techniques identify broad soundscape patterns that may not be obvious to human observers, though the black-box effect limits the ease of interpretation of such patterns. Therefore, pairing machine learning driven analysis with manual detection of fish calls could facilitate both broad interpretations of habitat differences and high-resolution signal-specific data.

In this study, we sought to establish if a reef undergoing coral restoration exhibited a detectable difference in its soundscape relative to nearby healthy and degraded reference reefs. We hypothesised that, due to ongoing reef community shifts through the coral restoration process, the restoration site would exhibit differences relative to the soundscapes of healthy and degraded sites. We used two analysis methods; first we manually detected and quantified unidentified fish sounds in the low frequency bandwidth. We compared this 'manual' analysis with machine learning led analysis which used a P-CNN to extract feature embeddings and unsupervised learning to visualise patterns within the low frequency soundscape using these embeddings. We developed the following hypotheses:

1. The frequency and diversity of manually detected fish sounds would be lowest in the degraded site, intermediate in the restoration site, and highest in the healthy reference site. This pattern would mirror the relative levels of coral cover in each site; we expected that differences in coral cover and associated fish communities would lead to an increase in the number of fish sounds present.

2. The variability in P-CNN derived feature embeddings would reflect site differentiation and the spectrum of increasing coral cover. We hypothesised that a P-CNN would detect similar differences to those observed by manually listening to fish sounds.

## Methods

### Site selection

We carried out this study in the Cousin Island Special Reserve, Seychelles (−4.33 S, 55.66 E), which hosts a coral restoration project run by Nature Seychelles since 2010. We selected three sites that differed in terms of restoration intervention, coral cover and diversity (Fig 1). Outplant site (−4.3333 S, 55.6576 E) is undergoing active restoration, and Degraded (−4.3346 S, 55.6583 E) and Healthy sites (−4.33608 S, 55.6590 E) represent the reference reefs (controls) where no intervention has occurred. The Cousin Island Special Reserve is protected from fishing and dive tourism, minimising direct, local anthropogenic impacts on the reef, although all three sites experienced mass bleaching in 1998 and 2016 [43] and minor bleaching in 2021 (Nature Seychelles unpublished data). The sites are rubble-dominated, with underlying granite structure. Each site has a depth range of 6–15 metres. The sites are around 50 metres apart from each other, meaning that abiotic conditions are consistent due to their proximity of location.

We refer to the Society for Ecological Restoration (SER) definition of "restoration" as any activity undertaken to assist recovery of a degraded ecosystem [44] and thus consider restoration as an ongoing process. All references to the restoration or Outplant site refer to a reef where active restoration is practiced. Since 2012, the Outplant site has undergone regular, active restoration with ~15 cm long (maximum branch length) nursery-grown fragments of *Acropora spp.* and *Pocillopora spp.* being planted using a concrete adhesive at a density of 4–8 fragments per m². At the time of this study (November 2023), approximately half of the 16 by 80 square metre area had been planted with corals at this density

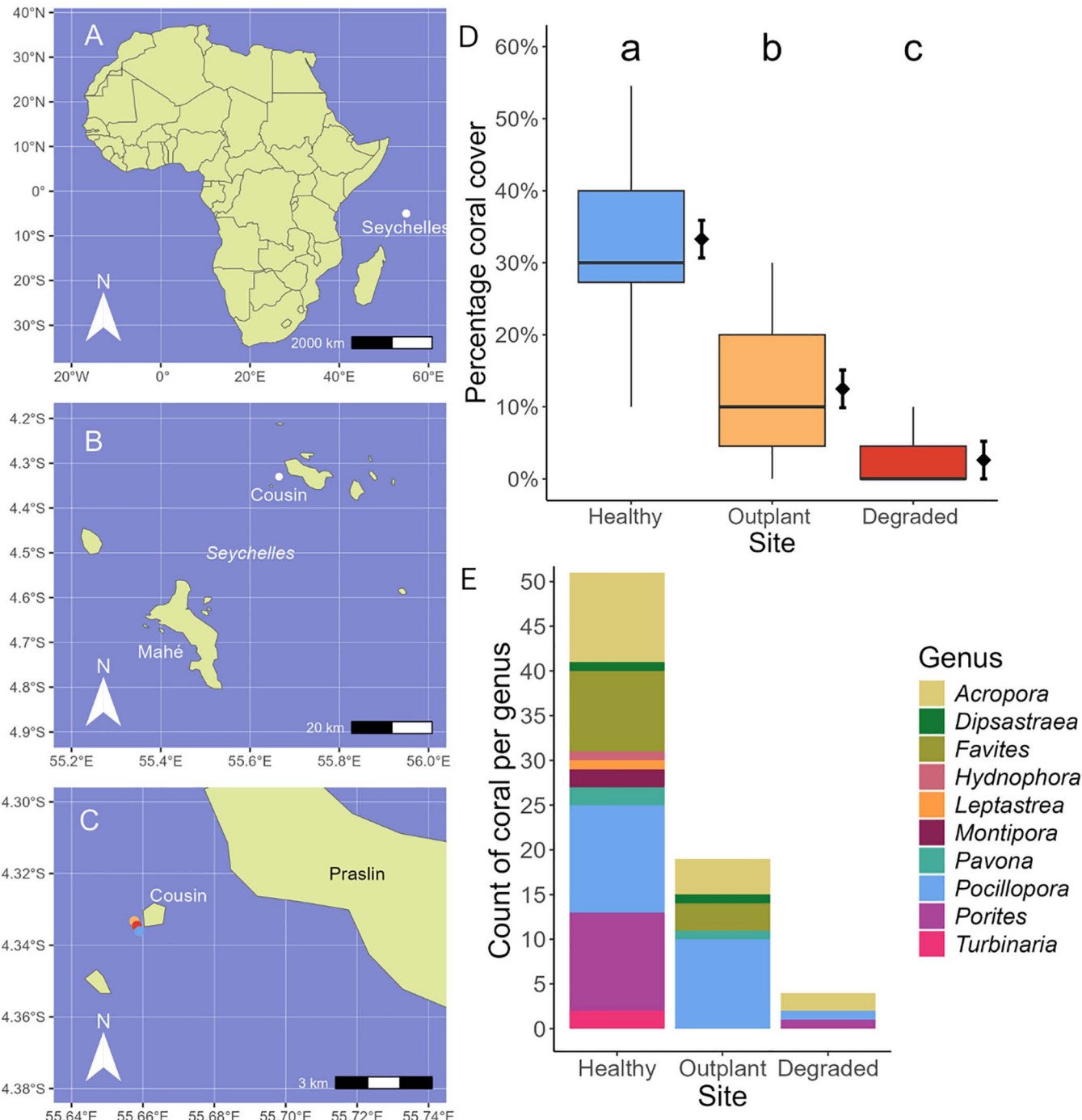

**Fig 1. Description of study sites at Cousin Island Special Reserve, Seychelles. A)** Position of the Seychelles, in the Western Indian Ocean. **B)** Seychelles inner islands region, study location (Cousin island) denoted by a white dot. **C)** Site location of the three reef sites included in this study. **D)** Coral cover (%) at Healthy, Outplant, and Degraded sites. Boxplot illustrates the median (middle line), interquartile range (coloured boxes), and full range (whiskers) of the data. Diamonds and error bars represent the model estimated mean coral cover and standard error (SE) at each site. Lower case letters refer to statistically significant differences between sites; for full statistical details, see Tables S1 and S2 in S1 File. **E)** Composition of coral genera at Healthy, Outplant, and Degraded reef sites. Count of coral per genus refers to the frequency of each coral genus recorded from PITs, relative to the total frequency of coral category observations within each PIT. Maps in 1A, 1B, and 1C were made with Natural Earth.

(Nature Seychelles unpublished data), although the bleaching event in 2016 resulted in a loss of coral cover from 9% in 2016 to 3% in 2018 (Nature Seychelles unpublished data). For full details of restoration process in Cousin Island Special Reserve refer to Frias-Torres et al. (2015) [45].

In November 2023 we collected three Point Intercept Transects (PIT) per site: each transect was 25 metres in length, transects were positioned parallel to shore and captured the full depth range (6–15 metres). Data were collected at 0.5 metre intervals over 25 metres. Coral cover and genus richness were calculated in smaller sub-transects of 5 metres length to increase replication and statistical reliability.

At the Healthy reference site, coral cover was highest (33%), followed by Outplant (12%), and Degraded (3%). For statistical comparisons, see Fig 1, Tables S1 and S2 in S1 File. Genus richness was greatest at the Healthy reference site (G = 10), followed by Outplant (G = 5) and Degraded (G = 3). The names 'Healthy' and 'Degraded' refer to these sites' ability to recover ecological function post-disturbance [46], based on baseline data collected prior to the mass bleaching events of 1998 and 2016 [47,48], and ongoing monitoring [49–51].

## Acoustic sampling design

We sampled reef soundscapes throughout a full lunar cycle in November 2023, recording over a 24-hour period every week, corresponding to the four lunar quarters. We defined lunar quarters as the night preceding and the night of the peak of each lunar phase. In the sampling design we included the balanced use of two HydroMoth recorders (www.openacous-ticdevices.info) at each site, to ensure that device ID did not interfere with site comparisons, and alternated which site was recorded at the peak night of the phase (Table S3 in S1 File). The HydroMoth device is a low-cost hydrophone with in-built battery and memory storage capability [14]. We placed recorders at the centre of each site, at a consistent depth of 8–9 m, 20 cm above the substrate, and attached to a concrete block.

We recorded over a frequency bandwidth of 0-48kHz to capture the entire soundscape. The HydroMoth device comes with a range of gain settings; we selected the medium setting – level 3 – based on pilot recordings. This setting allowed us to capture fish vocalisations without risk of clipping. We selected a recording: sleep duty cycle of 1:4 minutes to ensure recording length exceeded the maximum duration of most fish vocalisations [13], to gain a representative sample through time periods, and to maximise efficiency of battery use and recording time. We recorded, using this duty cycle, for a period of 60 minutes at four times per day: sunrise (05:30–06:30), midday (11:30–12:30), sunset (17:30–18:30), and midnight (23:30–00:30). We elected to sample for one hour per time phase to target maximum potential variability between time phases. These sampling times were kept consistent throughout the study period; sunrise time ranged from 05:52–05:55, and sunset time ranged from 18:08–18:16. This resulted in twelve 1-minute long recordings for each of the four 1-hour time periods per 24-hour sampling period per week, resulting in a total of 576 recordings over the entire study period.

Prior to analysis, we visually screened all recordings using Audacity ® Version 3.4.2 (www.audacityteam.org) and removed any occurrences of anthropogenic sound (for visual examples see Figure S1 in S1 File). Three recordings (0.005% of the total) contained anthropogenic sound and were omitted from the subsequent steps. One sampling period (12 recordings) at the Outplant site was contaminated with tidal flow sound during a particularly strong New Moon tide; this recording was also removed from analysis.

Two approaches governed the acoustic analysis phase (Fig 2). Firstly, we manually counted unidentified fish sounds to quantify their abundance and richness. Secondly, we performed feature extraction using a pre-trained convolutional neural network (P-CNN), SurfPerch, which is optimised for extracting informative feature embeddings from coral reef audio [52]. Before beginning either approach, we applied a bandwidth filter targeting the low-frequency range (0–1 kHz) typically dominated by fish sounds [11,13]. These approaches work in complement: manual detection focuses on a single taxon in detail, while the machine learning approach operates as a black box and does not have a pre-determined target taxon, but rather detects broad scale patterns in the low frequency band soundscape [53].

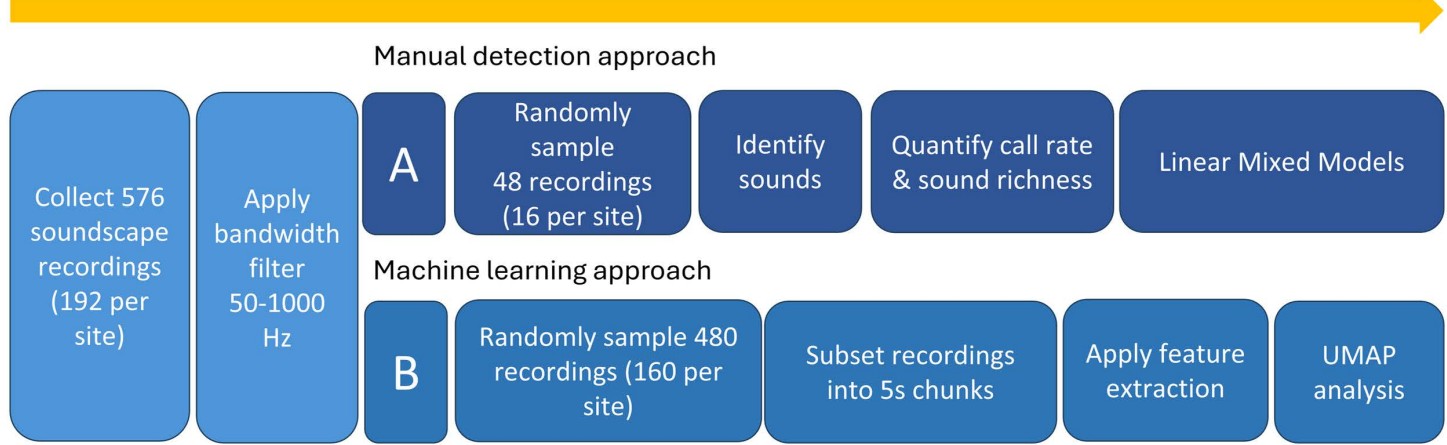

**Fig 2. Acoustic Analysis Workflow diagrams of the two approaches taken to analyse soundscape data, a manual detection (A) and machine learning approach (B).**

[A] Manual detection of call rates and sound richness

For manual analysis (human observers listening to soundscapes and visually checking spectrograms), we randomly selected one recording from each sample period, creating a subset of 48 recordings. A single observer (E.M.C.) then listened to all of the recordings, and inspected the spectrograms in Audacity. While acknowledging that a single observer may limit the generalisability of sound types, we elected to proceed with a single observer for the benefits of controlling for differences in observer, and employed a conservative definition of sound type, based on a minimum number of observations (n = 3) and grouping any similar sounds. The observer recognised 15 distinct fish call types, which they described in terms of pitch, rhythm and duration (Table S4 in S1 File). To quantify the call rate and sound richness per recording, the observer noted the frequency of occurrence of each of these sound types in each recording. We identified the individual vocalisations as likely to be produced by fishes based on their similarity to calls recorded by previous studies [11,13,54] (Table S4 in S1 File). We selected call rate and sound richness as relevant metrics based on their recent use in published work [11,27].

We compared call rates between sites with a Linear Mixed Model (LMM) from the lme4 package (v1.1-36) [55] as conditions for normality were met (Shapiro-Wilks test). We calculated sound richness with the vegan package (v2.6-4) [56] and compared between sites with a Generalised Linear Mixed Effects model (GLMM): we employed a Gamma distribution as the data was non-negative, continuous, and positively skewed. In both models, we included time of day as a fixed effect. In the call rate model, we also included lunar phase as a fixed effect; in the sound richness model lunar phase did not improve model fit so it was excluded for parsimony and interpretability. We assessed model fit through visual inspection of residual plots using the DHARMa package (v0.4.7) [57]. We evaluated models by comparison with a null model (ANOVA), and using the emmeans package (v1.10.7) [58] to test interactions between sites, lunar phase, and time of day. Throughout we considered p-values significant at the 95% confidence level (p ≤ 0.05). All statistical analysis and figure creation was carried out in *R* v4.3.3 [59] using the Tidyverse packages (v2.0.0) [60].

[B] Machine Learning approach

Of the original 576 recordings, inclusive of the recordings used for manual analysis, we used 480 one-minute recordings for the machine learning approach. We randomly sampled 10 out of every 12 one-minute long recordings as this allowed for an even number of recordings per timepoint. After removal of anthrophony and geophony, some timepoints only had 10 or 11 recordings remaining; therefore this downsampling of all timepoints to 10 recordings ensured equal sample

sizes across the dataset (Fig 2). In this random selection, 38 of the 48 recordings manually analysed were included in the machine learning analysis. We divided each of these one-minute recordings into twelve five-second samples; the final sample size was therefore 5760 recordings each of 5 seconds length. We used SurfPerch, a Pretrained Convolutional Neural Network (P-CNN) trained and evaluated on a diverse set of coral reef data gathered from around the tropics [52]. SurfPerch computes the log-mel spectrogram from each sample and then extracts a 1280 feature embedding from this.

We input the resultant feature embeddings into the Uniform Manifold Approximation and Projection (UMAP) algorithm for dimensionality reduction and visualisation [61]. UMAP is an unsupervised machine learning algorithm that is designed to project high-dimensional data, such as feature embeddings, into a lower-dimensional space while preserving both global and local structure as much as possible [61]. UMAP generated the two axes of greatest variation using the following parameters: nearest neighbours = 40, minimum distance = 0.3, and dimensions = 2. We visually inspected the resultant plots. We performed all aspects of the machine learning approach in Google Colab (www.colab.research.google.com; [62]).

## Results

### Manual detection of call rates and sound richness

Site had a large effect on call rate, both in comparison of Healthy and Degraded sites (d = −1.86, lower CL = −2.70, upper CL = −1.03) and Outplant and Degraded sites (d = −1.05, lower CL = −1.80, upper CL = −0.29) (Table S5 in S1 File). The mean call rate in each recording was more than 50% greater at the Healthy (8.5 ± 0.54) and Outplant (6.75 ± 0.54) sites relative to the Degraded site (4.5 ± 0.54) (Fig 3A, Table S5 in S1 File). The Outplant site exhibited 20% fewer calls per recording than the Healthy site but this difference was not significant (Fig 3A, Table S6 in S1 File). Sound richness of Outplant recordings did not differ from either Healthy or Degraded soundscapes, though richness of Degraded recordings was significantly less than Healthy (Fig 3B, Tables S7 and S8 in S1 File).

### Machine Learning approach

The UMAP plots produced two distinct groupings, one which exclusively contained samples from the Healthy site and another which primarily consisted of Outplant and Degraded site samples (Fig 4).

All recordings from Outplant and Degraded sites clustered together on the UMAP, (Fig 4). However, there were two different clusters both associated with Healthy recordings; the difference between these two clusters was driven by lunar phase. Full moon recordings from the Healthy site clustered with all Outplant and Degraded recordings, while Healthy recordings from other lunar phases clustered independently (Fig 5). There was no clustering pattern evident when samples were coloured by time of day (Figure S2 in S1 File).

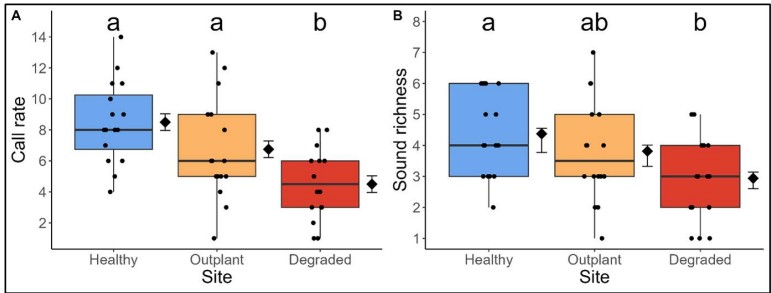

**Fig 3. Fish sounds differ in abundance and richness by site. A)** The effect of site on call rate (the total number of fish calls detected per recording, irrespective of sound type). **B)** The effect of site on sound richness (the number of different fish call types detected in each recording). Diamonds and error bars represent the model estimated mean and standard error (SE). Scattered dots demonstrate the spread of raw data. Lower case letters denote statistically significant differences.

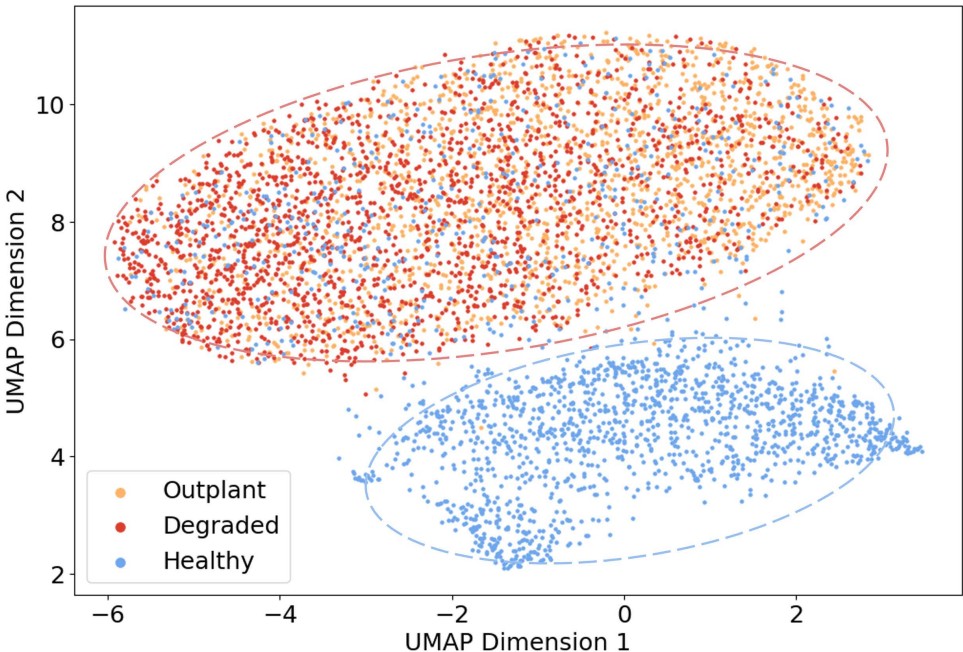

**Fig 4. UMAP visualisation of soundscape clustering by site.** UMAP Dimensions 1 and 2 refer to the axes of greatest variation. Each point represents a 5-second audio sample from a recording, the colours relate to the three reef sites: Healthy, Outplant, and Degraded.

## Discussion

We sought to establish whether soundscapes differed between a restoration site (Outplant) and adjacent Degraded and Healthy reference reefs. We employed manual detection and identification of call rates and sound richness, and machine learning driven UMAP analysis. The extent to which Outplant soundscapes were classified as similar to either Healthy or Degraded soundscapes differed depending on the analysis method used. Manual detection of fish calls classified Outplant soundscapes as similar to Healthy reefs in terms of call rate and sound richness. By contrast, machine learning analysis classified Outplant and Degraded soundscapes as similar, with Healthy soundscapes differing except during the Full Moon phase. We discuss the possibility that these results indicate that the restored site is exhibiting a recovery-stage soundscape, and reiterate the value of this combined PAM analysis approach (manual and machine learning) rather than any single analysis type.

The abundance and richness of manually detected fish calls was similar in Healthy and Outplant soundscapes, which may reflect a similarity in their respective soniferous fish communities (Fig 3). In contrast, fewer fish calls were recorded at the Degraded site, which could be indicative of either lower abundance of fish and/or less vocal activity by fishes. Acoustic activity in fishes has previously been described as a behavioural response, for instance to predator presence [63,64], therefore this result cannot necessarily be directly ascribed to lower abundance of fishes. Moreover, sound richness (the number of different sound types present in recordings) was lowest in Degraded soundscapes; this likely indicates that the richness of sound-producing taxa was lower in this site – although we acknowledge that the same fish species may produce a range of sound types, meaning that different sound types may not always indicate different species [65,66]. Identification of fish sounds to genus or species level would provide additional information as to the community and their contribution to reef function: this would be an effective avenue of future work to develop detailed indicators of reef recovery. We infer from these results that the similarity of Outplant and Healthy soundscapes reflects similar biological communities in terms of the abundance, richness, and vocal activity of fishes.

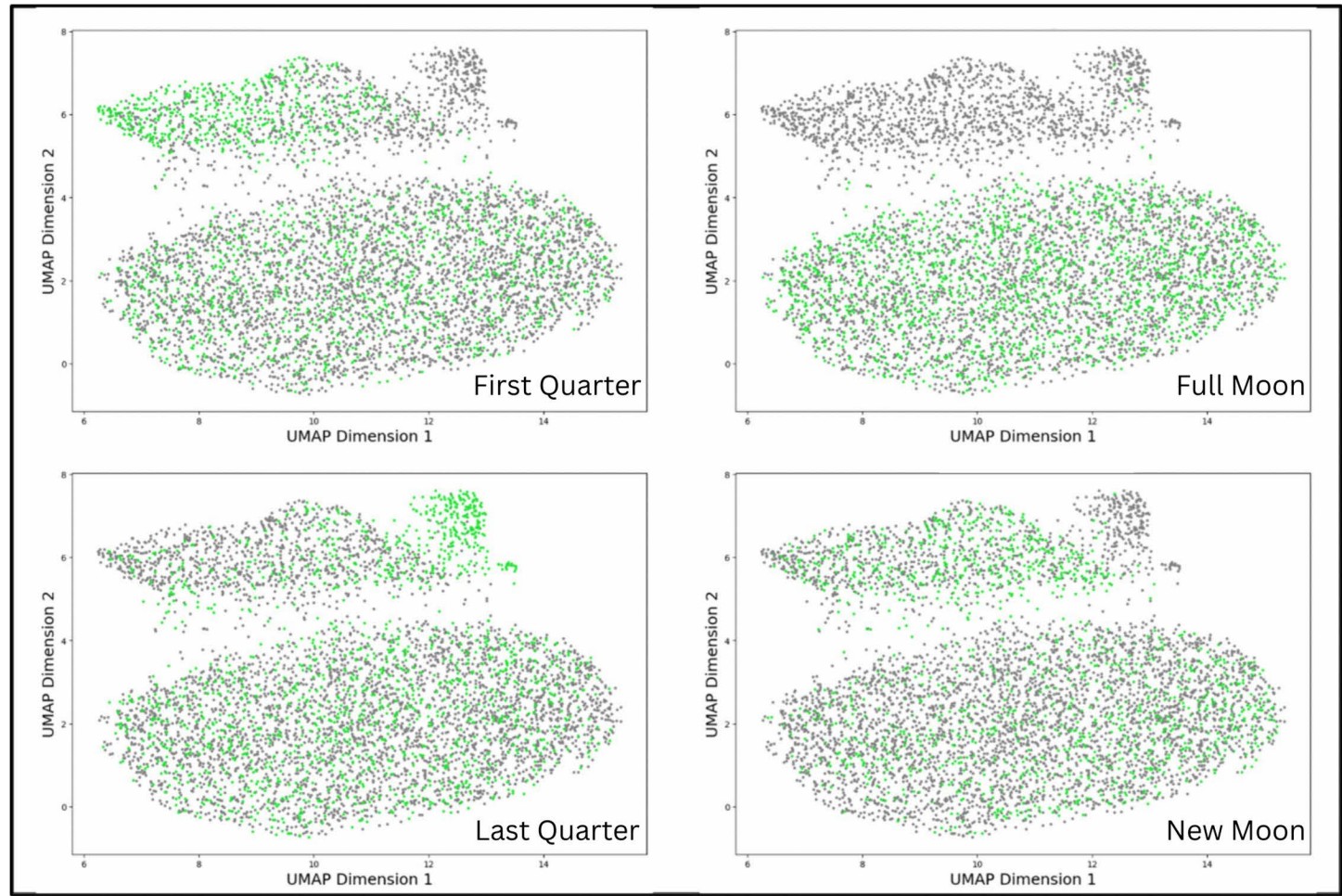

**Fig 5. UMAP visualisation of soundscape clustering by lunar phase.** UMAP Dimensions 1 and 2 refer to the axes of greatest variation. Each point represents a 5-second audio sample from a recording. In each panel, one lunar phase is highlighted in green, the other three phases are grey.

In agreement with the manual analysis, UMAP analysis revealed differences between the soundscapes of the Healthy and Degraded sites, illustrating the ability of this technique to differentiate between reefs of different quality in terms of coral cover (Fig 4). The grouping of Outplant and Degraded recordings into a single cluster therefore indicates that the recovery in soundscape observed in terms of fish calls is not yet reflected in this analysis of the broader soundscape. This finding may reflect the relatively narrow difference (~10%) in coral cover (Fig 1) between Outplant and Degraded sites, though greater site replication over this range of coral cover would be necessary to confirm this result.

Moreover, the UMAP classification of Outplant and Degraded sites as highly similar indicates an alternative driver of soundscape patterns than we detected in the abundance and richness of fish calls. UMAP analysis revealed variation within the Healthy soundscape according to lunar phase (Fig 5). This result suggests that the factors driving clustering are also associated with lunar variation. Whilst the P-CNN was optimised to identify features relevant to fish and other coral reef sounds, it still represents a black-box tool, meaning we do not know exactly which elements of the soundscape drove the UMAP groupings. Figure S4 in S1 File shows that manually analysed recordings were evenly distributed throughout the UMAP groupings, which confirms that the manual and machine learning analyses highlight different aspects of the soundscape. There are several possible drivers of the broader soundscape patterns characterised by the UMAP analysis.

One plausible driver of the broader soundscape patterns that were characterised by the UMAP analysis is invertebrate sounds, since invertebrate sounds typically vary in intensity with the lunar cycle [31]. Although these sounds are unlikely to dominate in the filtered low frequency bandwidth, the consistency of this result irrespective of bandwidth filter (Figure S3 in S1 File) implies that these patterns are present throughout the soundscape. Alternatively, in line with previous work we have discussed fish vocalisations in terms of their call rate and sound richness [11,27,33], but it is possible that the P-CNN identified features based on their pitch, volume, and/or complexity (e.g., number of concurrent sounds). Elucidating the ecological drivers of machine learning analyses is an important direction for future work to limit the black-box effect and facilitate stronger, ecologically relevant interpretations. Nonetheless, results derived from current machine-learning driven methods represent an improvement on acoustic indices in terms of accuracy, nuance, and generalisability [38]. Acoustic indices were not included in this work due to their inconsistency in detecting site differences, and the difficulty of ecologically meaningful interpretation [67,68].

The Outplant site is undergoing active restoration, which implies an ongoing community shift from a degraded towards a healthier state. With this in mind, we interpret the results of both analyses collectively as suggestive of a shifting community. Just as restored reefs often comprise ecologically novel communities that differ from both healthy and degraded reef states [69,70], we suggest that soundscapes on restored reefs are likely to exhibit characteristics that are distinct from both healthy and degraded soundscapes. For instance, the abundance and richness of fish calls at the Outplant site may indicate early-stage recovery, but the machine-learning clustering of these soundscapes alongside degraded soundscapes indicates that a broader community shift across other taxonomic groups has not yet occurred. Although this study was limited to a single restoration site at a single timepoint, future work could include multiple restoration sites with a range of coral cover through time to further quantify the differences in soundscapes between restored reefs at different successional timepoints. We posit the idea that restored reefs might have soundscapes that are specific to different points in the restoration process, with different elements of the reef soundscape recovering at different stages of restoration. This might lead to conflicting descriptions of soundscape activity according to taxa or methodology – although we acknowledge that further replication would be needed to fully test this hypothesis. Collectively, these findings highlight the relevance of PAM as a measure of restoration progress. In complement with traditional visual surveys, PAM provides critical nuance on the changes to reef function resulting from coral restoration work. This furthers our understanding of variability in recovery times of different reef functions in response to coral restoration, and reduces reliance on metrics focused on single taxa [1,4,5,70]. We have demonstrated the need for mixed-method analysis to facilitate discussion of soundscape recovery at the scale of the broader soniferous community, particularly to avoid over- or under-estimations of reef recovery based on single taxa metrics.

We detected acoustic differences between an active restoration site and reference reefs, indicating some recovery of the soundscape. Indications of recovery were limited to fish calls, while broader soundscape patterns categorised by machine learning highlighted the similarity of restoration and degraded sites. The use of multiple manual and machine learning analyses in this study provides an accessible framework for restoration practitioners to develop this monitoring tool further. We echo demands for future work to focus on monitoring and conserving reef function and have provided evidence of a tool that can contribute to measuring the state of coral reef communities.

## Suppo rting information

**S1 File. This file contains Tables S1-9 and Figures S1-5.**
(DOCX)

## Acknowledgments

We would like to thank Aylisa Joubert, Jacques Aglae, and Morgan Rose for their assistance in the field. Many thanks to Verena Schoepf for the early advice and mentoring. The authors would like to thank Nature Seychelles, through whom this study was conducted, particularly Sam Ramkalawan and Kerstin Henri, who manage the reef restoration project. We

would like to thank UNDP Mauritius, the Seychelles Government and the Ministry of Agriculture, Climate Change and Environment for their support.

## Author contributions

**Conceptualization:** Emily Maria Croasdale, Luca Saponari, Timothy A. C. Lamont.

**Data curation:** Emily Maria Croasdale.

**Formal analysis:** Emily Maria Croasdale.

**Funding acquisition:** Luca Saponari, Charlotte Dale, Nirmal Shah.

**Investigation:** Emily Maria Croasdale, Luca Saponari, Charlotte Dale.

**Methodology:** Emily Maria Croasdale, Ben Williams, Timothy A. C. Lamont.

**Project administration:** Charlotte Dale, Nirmal Shah.

**Resources:** Luca Saponari, Nirmal Shah, Timothy A. C. Lamont.

**Software:** Emily Maria Croasdale, Ben Williams.

**Supervision:** Luca Saponari, Timothy A. C. Lamont.

**Visualization:** Emily Maria Croasdale, Timothy A. C. Lamont.

**Writing – original draft:** Emily Maria Croasdale.

**Writing – review & editing:** Emily Maria Croasdale, Luca Saponari, Charlotte Dale, Ben Williams, Timothy A. C. Lamont.

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
