## [Decision Letter · Decision Letter 0]

14 Sep 2025

PONE-D-25-26036Coral restoration alters reef soundscapes but machine learning and manual analyses suggest different recovery ratesPLOS ONE

Dear Dr. Croasdale,

Thank you for submitting your manuscript to PLOS ONE. After careful consideration, we feel that it has merit but does not fully meet PLOS ONE’s publication criteria as it currently stands. Therefore, we invite you to submit a revised version of the manuscript that addresses the points raised during the review process.

If applicable, we recommend that you deposit your laboratory protocols in protocols.io to enhance the reproducibility of your results. Protocols.io assigns your protocol its own identifier (DOI) so that it can be cited independently in the future. For instructions see: https://journals.plos.org/plosone/s/submission-guidelines#loc-laboratory-protocols. Additionally, PLOS ONE offers an option for publishing peer-reviewed Lab Protocol articles, which describe protocols hosted on protocols.io. Read more information on sharing protocols at . Additionally, PLOS ONE offers an option for publishing peer-reviewed Lab Protocol articles, which describe protocols hosted on protocols.io. Read more information on sharing protocols at https://plos.org/protocols?utm_medium=editorial-email&utm_source=authorletters&utm_campaign=protocols..

We look forward to receiving your revised manuscript.

Kind regards,

Parviz Tavakoli-Kolour

Academic Editor

PLOS ONE

**Journal Requirements:**

1. When submitting your revision, we need you to address these additional requirements. Please ensure that your manuscript meets PLOS ONE's style requirements, including those for file naming. The PLOS ONE style templates can be found at https://journals.plos.org/plosone/s/file?id=wjVg/PLOSOne_formatting_sample_main_body.pdf and https://journals.plos.org/plosone/s/file?id=ba62/PLOSOne_formatting_sample_title_authors_affiliations.pdf 2. We note that the grant information you provided in the ‘Funding Information’ and ‘Financial Disclosure’ sections do not match.  When you resubmit, please ensure that you provide the correct grant numbers for the awards you received for your study in the ‘Funding Information’ section. 3. Thank you for stating the following in the Acknowledgments Section of your manuscript: We would like to thank Aylisa Joubert, Jacques Aglae, and Morgan Rose for their assistance in the field. Many thanks to Verena Schoepf for the early advice and mentoring. The authors would like to thank Nature Seychelles, through whom this study was conducted, particularly S. Ramkalawan and K. Henri, who manage the reef restoration project. We would like to thank the Seychelles Ministry of Agriculture, Climate Change and Environment, the Adaptation Fund, UNDP Mauritius and the Government of Seychelles for the project support. We note that you have provided funding information that is not currently declared in your Funding Statement. However, funding information should not appear in the Acknowledgments section or other areas of your manuscript. We will only publish funding information present in the Funding Statement section of the online submission form. Please remove any funding-related text from the manuscript and let us know how you would like to update your Funding Statement. Currently, your Funding Statement reads as follows: Sources of funding for this project include the Adaptation Fund of the United Nations Development Programme in Mauritius, awarded to Nature Seychelles (LS, CD, NS) for project PIMS 5736 (undp.org/mauritius-seychelles/projects/restoring-marine-ecosystem-services-rehabilitating-coral-reefs-meet-changing-climate-future), a Madeleine Julie Vervoort Fonds grant from the Amsterdams Universiteitsfonds (auf.nl) awarded to EMC, and a research fellowship from the 1851 Royal Commission, awarded to TACL (royalcommission1851.org). Funders played no role in any part of the study or manuscript preparation.  Please include your amended statements within your cover letter; we will change the online submission form on your behalf. 4. Please note that your Data Availability Statement is currently missing the DOI/accession number of each dataset OR a direct link to access each database. If your manuscript is accepted for publication, you will be asked to provide these details on a very short timeline. We therefore suggest that you provide this information now, though we will not hold up the peer review process if you are unable. 5. When completing the data availability statement of the submission form, you indicated that you will make your data available on acceptance. We strongly recommend all authors decide on a data sharing plan before acceptance, as the process can be lengthy and hold up publication timelines. Please note that, though access restrictions are acceptable now, your entire data will need to be made freely accessible if your manuscript is accepted for publication. This policy applies to all data except where public deposition would breach compliance with the protocol approved by your research ethics board. If you are unable to adhere to our open data policy, please kindly revise your statement to explain your reasoning and we will seek the editor's input on an exemption. Please be assured that, once you have provided your new statement, the assessment of your exemption will not hold up the peer review process. 6. We note that Figure 1 in your submission contain map/satellite images which may be copyrighted. All PLOS content is published under the Creative Commons Attribution License (CC BY 4.0), which means that the manuscript, images, and Supporting Information files will be freely available online, and any third party is permitted to access, download, copy, distribute, and use these materials in any way, even commercially, with proper attribution. For these reasons, we cannot publish previously copyrighted maps or satellite images created using proprietary data, such as Google software (Google Maps, Street View, and Earth). For more information, see our copyright guidelines: http://journals.plos.org/plosone/s/licenses-and-copyright. We require you to either present written permission from the copyright holder to publish these figures specifically under the CC BY 4.0 license, or remove the figures from your submission: a. You may seek permission from the original copyright holder of Figure 1 to publish the content specifically under the CC BY 4.0 license.   We recommend that you contact the original copyright holder with the Content Permission Form (http://journals.plos.org/plosone/s/file?id=7c09/content-permission-form.pdf) and the following text:“I request permission for the open-access journal PLOS ONE to publish XXX under the Creative Commons Attribution License (CCAL) CC BY 4.0 (http://creativecommons.org/licenses/by/4.0/). Please be aware that this license allows unrestricted use and distribution, even commercially, by third parties. Please reply and provide explicit written permission to publish XXX under a CC BY license and complete the attached form.” Please upload the completed Content Permission Form or other proof of granted permissions as an "Other" file with your submission. In the figure caption of the copyrighted figure, please include the following text: “Reprinted from [ref] under a CC BY license, with permission from [name of publisher], original copyright [original copyright year].” b. If you are unable to obtain permission from the original copyright holder to publish these figures under the CC BY 4.0 license or if the copyright holder’s requirements are incompatible with the CC BY 4.0 license, please either i) remove the figure or ii) supply a replacement figure that complies with the CC BY 4.0 license. Please check copyright information on all replacement figures and update the figure caption with source information. If applicable, please specify in the figure caption text when a figure is similar but not identical to the original image and is therefore for illustrative purposes only.The following resources for replacing copyrighted map figures may be helpful: USGS National Map Viewer (public domain): http://viewer.nationalmap.gov/viewer/The Gateway to Astronaut Photography of Earth (public domain): http://eol.jsc.nasa.gov/sseop/clickmap/Maps at the CIA (public domain): https://www.cia.gov/library/publications/the-world-factbook/index.html and https://www.cia.gov/library/publications/cia-maps-publications/index.htmlNASA Earth Observatory (public domain): http://earthobservatory.nasa.gov/Landsat:
http://landsat.visibleearth.nasa.gov/USGS EROS (Earth Resources Observatory and Science (EROS) Center) (public domain): http://eros.usgs.gov/#Natural Earth (public domain): http://www.naturalearthdata.com/ 7. If the reviewer comments include a recommendation to cite specific previously published works, please review and evaluate these publications to determine whether they are relevant and should be cited. There is no requirement to cite these works unless the editor has indicated otherwise.

Reviewers' comments:

Reviewer's Responses to Questions

**Comments to the Author**

1. Is the manuscript technically sound, and do the data support the conclusions?

Reviewer #1: Yes

Reviewer #2: Partly

2. Has the statistical analysis been performed appropriately and rigorously? 

Reviewer #1: Yes

Reviewer #2: Yes

3. Have the authors made all data underlying the findings in their manuscript fully available?

Reviewer #1: Yes

Reviewer #2: No

4. Is the manuscript presented in an intelligible fashion and written in standard English?

Reviewer #1: Yes

Reviewer #2: Yes

5. Review Comments to the Author

**Reviewer #1:** a) The conceptual link between reef function, ecological health, and soundscape recovery should be more clearly articulated. A concise framework or figure would help clarify the interpretation of PAM data as a proxy for ecosystem function.a) The conceptual link between reef function, ecological health, and soundscape recovery should be more clearly articulated. A concise framework or figure would help clarify the interpretation of PAM data as a proxy for ecosystem function.

b) The hypotheses could be stated more explicitly, outlining a priori expectations for both manual and machine learning analyses.

c) The use of a single restoration site should be acknowledged as a limitation affecting generalisability. Clarify whether abiotic factors were controlled across sites.

d) Justify choices in acoustic sampling design, such as the 1:4 duty cycle and selection of 10/12 recordings per phase.

e) Manual analysis by a single observer should be discussed in terms of potential bias, and whether standardised resources were used for fish call classification.

f) UMAP outputs are informative but difficult to interpret ecologically. Consider discussing potential drivers of clustering and whether acoustic indices (e.g., ACI, ADI) were considered.

g) The term “restoration-specific soundscape” is compelling but needs a clearer functional definition.

h) Expand briefly on the conservation implications of PAM-based approaches and their integration into reef monitoring.

i) Ensure consistency in reference formatting and check for updated versions of any preprint citations.

**Reviewer #2:** I would like to thank the authors for their effort in preparing the manuscript “Coral restoration alters reef soundscapes but machine learning and manual analyses suggest different recovery rates”. The manuscript raises a relevant topic with potential value to the field. However, I believe that in its current state, it does not provide sufficient evidence to support the claims. The described statistical analysis and machine learning approach are sound and detailed enough to be understood. Nonetheless, I believe the authors should consider adding supportive evidence or expanding the main results to strengthen their claims. I would like to thank the authors for their effort in preparing the manuscript “Coral restoration alters reef soundscapes but machine learning and manual analyses suggest different recovery rates”. The manuscript raises a relevant topic with potential value to the field. However, I believe that in its current state, it does not provide sufficient evidence to support the claims. The described statistical analysis and machine learning approach are sound and detailed enough to be understood. Nonetheless, I believe the authors should consider adding supportive evidence or expanding the main results to strengthen their claims.

L194–195. The authors state that a bandwidth filter for the low-frequency range, which typically relates to fish sounds, was applied before being used in both manual and ML analyses.

L318–321. The authors suggest that it is plausible to believe that the broader soundscape patterns seen in UMAP are due to invertebrate sounds. Do the authors mean that invertebrate sounds were also filtered in the analysis? Could you provide some evidence? It was not clear to me whether the manual analysis was focusing on a single taxon and the ML on multiple taxa.

L206–207. “We identified the individual vocalizations as likely to be produced by fishes based on their similarity to calls recorded by previous studies (11,13,49).” Could the authors provide a list with the potential matches to the previous studies? It would be useful to see, at least in the supplementary material, all the fishes that were recognized. I believe this would strengthen your claims and add to the discussion.

L276–286. I understand how demanding the manual analysis can be, but could the different results between manual and ML be a matter of sample size? Would it be possible to run the ML analysis only using the same samples as in the manual analysis to see if you get similar results? Were the same samples from the manual analysis also randomly selected and used in the ML? It would be interesting to see where the samples used in the manual analysis fall in the UMAP graphs. This could help clarify what the ML “black box” is considering.

L227. Each recording used in the ML approach was 5 seconds in length. I believe the manual approach used longer recordings. I would argue that this makes the approaches more comparative than complementary, which changes what the authors claim to achieve.

Fig 1E. The genera names should be italicized. I believe “Fungidae” is the family name. Also, the “Healthy” bar has only 10 colors, while the results indicate G = 11.

The manuscript requires stronger evidence for its claims, clarification on methodological differences, and improved presentational of results. I recommend a major revision, with the expectation that the authors can significantly improve the manuscript through additional explanation and incorporation of further supporting evidence.

6. PLOS authors have the option to publish the peer review history of their article (what does this mean?). If published, this will include your full peer review and any attached files.). If published, this will include your full peer review and any attached files.

.

Reviewer #1: **Yes:** Musfera JahanMusfera Jahan

Reviewer #2: No

While revising your submission, please upload your figure files to the Preflight Analysis and Conversion Engine (PACE) digital diagnostic tool, https://pacev2.apexcovantage.com/. PACE helps ensure that figures meet PLOS requirements. To use PACE, you must first register as a user. Registration is free. Then, login and navigate to the UPLOAD tab, where you will find detailed instructions on how to use the tool. If you encounter any issues or have any questions when using PACE, please email PLOS at . PACE helps ensure that figures meet PLOS requirements. To use PACE, you must first register as a user. Registration is free. Then, login and navigate to the UPLOAD tab, where you will find detailed instructions on how to use the tool. If you encounter any issues or have any questions when using PACE, please email PLOS at figures@plos.org. Please note that Supporting Information files do not need this step.. Please note that Supporting Information files do not need this step.

---

## [Author Response · Author response to Decision Letter 1]

24 Mar 2026

We would like to thank both of the reviewers for taking the time to give us constructive feedback on this work, which we feel has greatly improved the manuscript. We have addressed the comments in full in the "response to reviewers" document.

---

## [Editor Report · Decision Letter 1]

1 Apr 2026

Coral restoration alters reef soundscapes but machine learning and manual analyses suggest different recovery rates

PONE-D-25-26036R1

Dear Dr. Croasdale,

We’re pleased to inform you that your manuscript has been judged scientifically suitable for publication and will be formally accepted for publication once it meets all outstanding technical requirements.

An invoice will be generated when your article is formally accepted. Please note, if your institution has a publishing partnership with PLOS and your article meets the relevant criteria, all or part of your publication costs will be covered. Please make sure your user information is up-to-date by logging into Editorial Manager at Editorial Manager® and clicking the ‘Update My Information' link at the top of the page. For questions related to billing, please contact  and clicking the ‘Update My Information' link at the top of the page. For questions related to billing, please contact billing support..

Kind regards,

Parviz Tavakoli-Kolour

Academic Editor

PLOS One

---

## [Editor Report · Acceptance letter]

PONE-D-25-26036R1

PLOS One

Dear Dr. Croasdale,

I'm pleased to inform you that your manuscript has been deemed suitable for publication in PLOS One. Congratulations! Your manuscript is now being handed over to our production team.

Kind regards,

on behalf of

Dr. Parviz Tavakoli-Kolour

Academic Editor

PLOS One